# Mapping the molecular landscape of *Lotus japonicus* nodule organogenesis through spatiotemporal transcriptomics

Keyi Ye [1,10] ✉, Fengjiao Bu [1,10], Liyuan Zhong [2,10], Zhaonian Dong[1,10], Zhaoxu Ma [1,3,10], Zhanpeng Tang[1], Yu Zhang[1,4], Xueyong Yang[5], Xun Xu [6], Ertao Wang [7], William J. Lucas[1,8], Sanwen Huang [1,9], Huan Liu [2,6] ✉ & Jianshu Zheng [1] ✉

Legumes acquire nitrogen-fixing ability by forming root nodules. Transferring this capability to more crops could reduce our reliance on nitrogen fertilizers, thereby decreasing environmental pollution and agricultural production costs. Nodule organogenesis is complex, and a comprehensive transcriptomic atlas is crucial for understanding the underlying molecular events. Here, we utilized spatial transcriptomics to investigate the development of nodules in the model legume, *Lotus japonicus*. Our investigation has identified the developmental trajectories of two critical regions within the nodule: the infection zone and peripheral tissues. We reveal the underlying biological processes and provide gene sets to achieve symbiosis and material exchange, two essential aspects of nodulation. Among the candidate regulatory genes, we illustrate that *LjNLP3*, a transcription factor belonging to the NIN-LIKE PROTEIN family, orchestrates the transition of nodules from the differentiation to maturation. In summary, our research advances our understanding of nodule organogenesis and provides valuable data for developing symbiotic nitrogen-fixing crops.

Symbiotic nitrogen fixation (SNF) is an intricate biological process, in which legumes host rhizobia to overcome nitrogen deficiency. The rhizobia are capable of converting atmospheric dinitrogen into ammonia, a form that is utilizable by legumes, in exchange for carbohydrates provided by the legumes. A specialized plant organ, named the root nodule, forms to achieve this purpose, and understanding the mechanism of nodule organogenesis has been a long-standing goal for scientists seeking to transform more crops to have nitrogen-fixing nodules[1,2].

Nodule organogenesis is complex, and a comprehensive transcriptomic atlas is essential to reveal the underlying biological processes and functional genes involved[3,4]. However, sampling nodules

[1]Shenzhen Branch, Guangdong Laboratory of Lingnan Modern Agriculture, Genome Analysis Laboratory of the Ministry of Agriculture and Rural Affairs, Agricultural Genomics Institute at Shenzhen, Chinese Academy of Agricultural Sciences, Shenzhen, Guangdong 518120, China. [2]BGI Research, Wuhan 430074, China. [3]National Key Laboratory of Crop Genetic Improvement and National Centre of Plant Gene Research (Wuhan), College of Life Science and Technology, Huazhong Agricultural University, Wuhan 430070, China. [4]School of Agriculture, Sun Yat-sen University, Shenzhen 518107, China. [5]State Key Laboratory of Vegetable Biobreeding, Institute of Vegetables and Flowers, Chinese Academy of Agricultural Sciences, Beijing 100081, China. [6]State Key Laboratory of Agricultural Genomics, BGI Research, Shenzhen 518083, China. [7]National Key Laboratory of Plant Molecular Genetics, CAS Center for Excellence in Molecular Plant Sciences, Institute of Plant Physiology and Ecology, SIBS, Chinese Academy of Sciences, Shanghai, China. [8]Department of Plant Biology, College of Biological Sciences, University of California, Davis, CA 95616, USA. [9]National Key Laboratory of Tropical Crop Breeding, Chinese Academy of Tropical Agricultural Sciences, Haikou, Hainan 571101, China. [10]These authors contributed equally: Keyi Ye, Fengjiao Bu, Liyuan Zhong, Zhaonian Dong, Zhaoxu Ma. ✉e-mail: yekeyi@caas.cn; liuhuan@genomics.cn; zhengjianshu@caas.cn

across development for bulk transcriptomics is difficult, due to their subterranean formation. Indeterminate nodules have continuous meristems that develop all tissue types in their mature state[5]. Emerging techniques, such as laser-capture microdissection coupled with RNA sequencing, enable the detection of transcriptomes in tissues at various developmental stages within a mature indeterminate nodule. Studies on indeterminate nodules from *Medicago truncatula* have advanced our understanding of nodulation, including the discovery that orthologs of Arabidopsis genes linked to the QC (quiescent center) potentially regulate nodule meristem development[6,7]. In contrast, research on the other essential nodule types, namely determinate nodules characterized by the absence of continuous meristems, is still lacking, which has significantly impeded our holistic understanding of nodulation[8,9].

Spatial transcriptomics allows for the creation of in situ transcriptomes that integrate gene-expression levels with location information[10]. The resulting transcriptomes can be analyzed by incorporating morphological and anatomical observations, which is particularly advantageous in plant research where distinct marker genes for annotating different tissue types are often scarce. Additionally, it enables visualization of individual samples, overcoming sampling challenges that hinder research on indeterminate nodules. This methodology, including the Stereo-seq technique developed by BGI, has been successfully applied to several plant species since its initial testing on Arabidopsis, demonstrating its reliability and effectiveness in plant research[11–13].

In this study, we have employed Stereo-seq to delve into the molecular development of determinate nodules from *Lotus japonicus*. Our investigation delineates the developmental trajectories of the infection zone and peripheral tissues, two critical regions responsible for symbiosis and material exchange, the focal aspects of SNF. Genes functioning at each stage are unveiled. Notably, several key regulatory genes demonstrate that *L. japonicus* and *M. truncatula* share similar developmental trajectories, despite their distinct nodule types, providing solid evidence for the monophyly of Fabaceae, an essential theory for developing novel nitrogen-fixing crops[14,15]. Particularly, we have investigated the role of LjNLP3 (NIN-LIKE PROTEIN 3) in transitioning nodules from differentiation to maturation, enhancing our understanding of this important family's involvement in nodulation[16]. Together, our study provides a comprehensive dataset for investigating nodule organogenesis, alongside valuable insights into understanding nodulation, informing future research and agricultural applications.

## Results

### Spatial transcriptomic atlas of *L. japonicus* nodule organogenesis

Comprehensive tissue sectioning assays were first performed to monitor the progression of nodule development. In combination with earlier studies[8,17,18], four stages (5, 6, 8 and 10 days post-inoculation, dpi), spanning from nodule primordia to mature nodule, were selected as being representatives to perform Stereo-seq (Supplementary Fig. 1a–e). Considering that nodule development is a consecutive process, whereas samples in spatial transcriptomes contain only one layer of cells, we thus collected many samples to cover different developmental states. After quality control, spatial transcriptomic data of 72 nodule samples were developed, covering 24,929 genes (~80% of the annotated genes in *L. japonicus* ecotype Gifu genome)[2]. We aggregated datasets from individual sections into bins for subsequent analyses[19]. The numbers of unique molecular identifiers (UMIs) and genes, per bin, were then calculated (Supplementary Fig. 2a, b). Here, we observed that bins associated with root nodules exhibited lower counts of UMIs and genes at 10 dpi compared to earlier stages, indicating a potential reduction in mRNA content during later developmental stages[19–21].

Unsupervised clustering was next performed on the 19,238 bins (Supplementary Fig. 2c–g) and 14 clusters were established that reflected good agreement with cellular anatomy (Fig. 1a–c; Supplementary Fig. 1f). Hundreds of marker genes were identified (Supplementary Fig. 2f) and, in combination with those earlier investigated (Fig. 1d), we successfully annotated the main clusters (Fig. 1a and Supplementary Fig. 1g). Among them, clusters 3–9 corresponded to root nodules and occupied over half of the total bins (Supplementary Fig. 2c), exhibiting developmental specificity (Fig. 1c). Based on previous studies[8,17,18], and our anatomical investigations, these bins were divided into three main regions (Fig. 1a and Supplementary Fig. 1g). Region I included clusters 5, 8, and 9, corresponding to the infection zone at different developmental stages. Region II included clusters 4 and 7, corresponding to nodule meristems and peripheral tissues, respectively. Region III included clusters 3 and 6, primarily corresponding to the early nodule cortex (root cortex surrounding the nodule primordium) and nodule outer cortex, respectively.

### Fulfilling symbiotic requirements results in distinction of the infection zone

We next performed a correlation analysis on these clusters to uncover the relationship among their respective tissues. Four clusters (clusters 10–13), with comparatively fewer bins (7.1% of total bins; Supplementary Fig. 2c), were likely a mixture, as they lacked developmental patterns and were scattered across the boundaries (Fig. 1b, c), due to insufficient resolution of present spatial transcriptomics[11,19]. Thereby, we performed correlation analysis on clusters 0-9 and divided them into four modules (Supplementary Fig. 3a). The results indicated that nodule outer cortex (cluster 6) and peripheral tissues (cluster 7) were similar to root tissues (clusters 0, 1, and 2), which is consistent with their morphological appearance (Supplementary Fig. 1c).

Clusters 3, 4, and 8, corresponding to the early nodule tissues (Fig. 1c), were in one module with a considerable number of shared marker genes (Supplementary Fig. 3b). GO enrichment analysis revealed significant enrichment of genes related to nucleosome and ribosome (Supplementary Fig. 3c), which replicate during mitosis[22,23], confirming that cell division was active in the early nodule. Among the only marker genes of cluster 8, we identified several investigated genes and orthologs related to infection events, including rhizobial signal receptors *LjSYMRK* and *LjEPR3* (Supplementary Fig. 3d). This implied that cluster 8 was involved in the preparation for infection, which aligned well with the previously defined pre-infection zone[6,7].

Clusters 9 and 5 corresponded to the differentiating and differentiated stages of the infection zone after infection. They were distinct from other tissues (Supplementary Fig. 3a). To investigate their specialties, we performed GO enrichment analyses on their marker genes. Cluster 9 showed a prominent enrichment in transmembrane transport (Supplementary Fig. 3e), including transporters known to participate in SNF (Supplementary Data 2)[24,25]. Cluster 5 represented when the nodule was capable of fixing nitrogen, enriching genes involved in nitrogen metabolism (Supplementary Fig. 3f), including GS-GOGAT cycle genes expressed in the nitrogen fixation zone (Supplementary Data 2)[26–28]. Leghemoglobins (*Lbs*) were also enriched (Supplementary Data 2), and are known to bind oxygen to maintain a hypoxic condition essential for the rhizobia to function[29,30]. As is well-known, SNF is a metabolic symbiosis built upon the trading of materials between the legume and rhizobium[2]. Our results emphasized that the infection zone serves as the central site for this symbiosis, and the fulfilment of symbiotic requirements distinguishes it from other tissues.

### Exploring infection zone development at the molecular level

To refine the developmental trajectories of the infection zone, clusters 5, 8, and 9 were re-clustered to distinguish their different states without interference from other tissues (Fig. 2a). Notably, the pre-infection

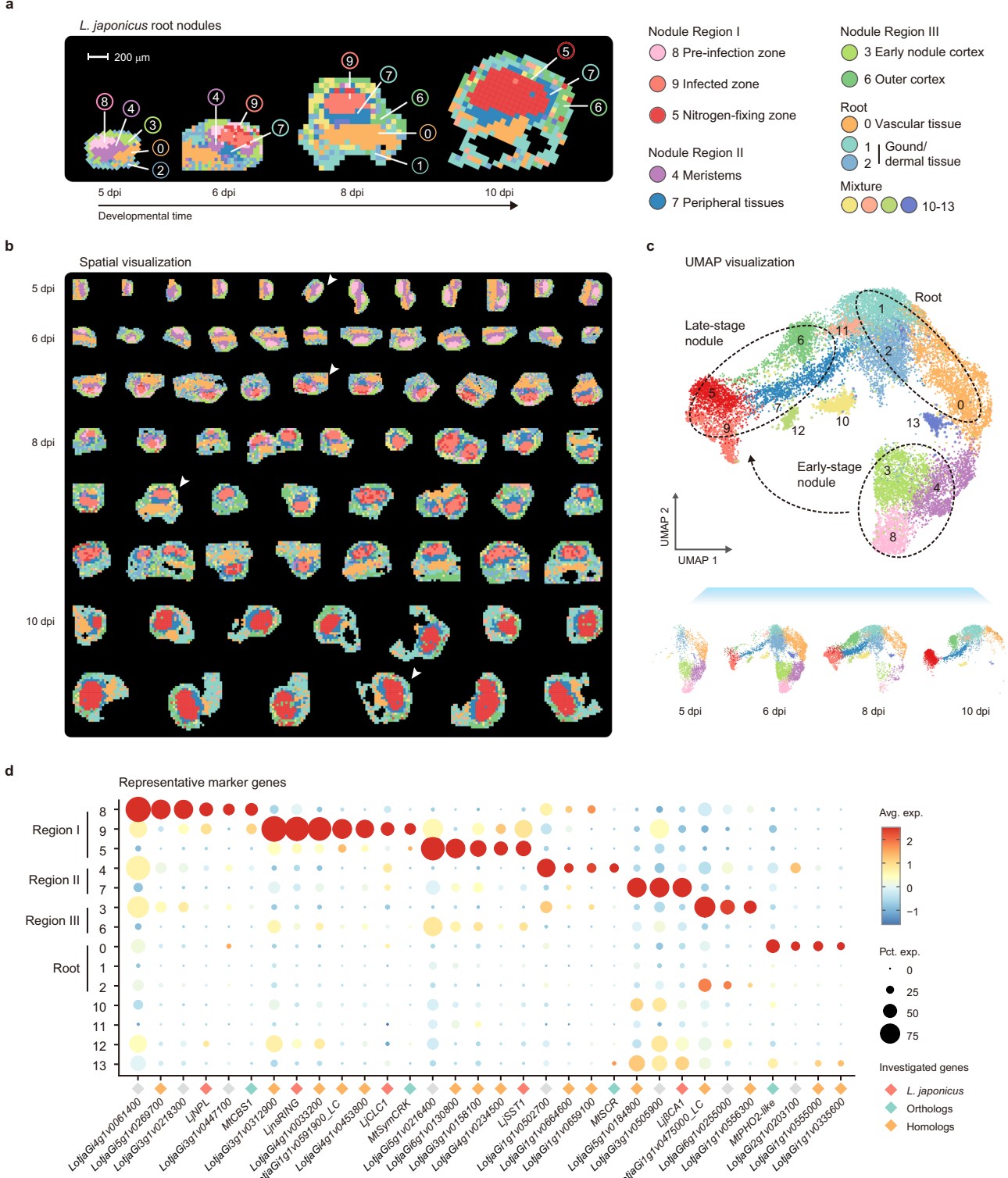

**Fig. 1 | Spatial transcriptomic atlas of *L. japonicus* nodule organogenesis. a** The spatially clustering results of representative samples at each nodule developmental stage. These samples are marked by white arrowheads in **b**, see below. Scale bar = 200 μm. The annotation on each cluster is listed on the right. Detailed anatomically-based annotation and sketch illustration are demonstrated in Supplementary Fig. 1f and 1g. **b** Spatial visualization depicting the clustering of 19,238 bins generated from 72 nodules. The samples are organized based on their developmental time and pseudotime in this study; samples from early to late states are arranged from top to bottom and left to right. The white arrowheads indicate the representative samples in **a**. **c** The UMAP (uniform manifold approximation and projection) dimension reduction graph depicting the clustering results. The main tissue types are circled and summarized. The arrowed black dotted line indicates the nodule developmental direction, based on the same UMAP plotted on the bottom that shows only bins from each stage. The full lists of marker genes for each cluster are provided in Supplementary Data 1. **d** Bubble plot showing the clustering expression pattern of selected marker genes for main tissue types. The clusters are briefly annotated on the left. Bubble size represents the percentage of bins expressing the indicated marker genes, and color encodes the average expression level. Investigated genes or homologs are marked, and detailed information on these marker genes is provided in Supplementary Data 1.

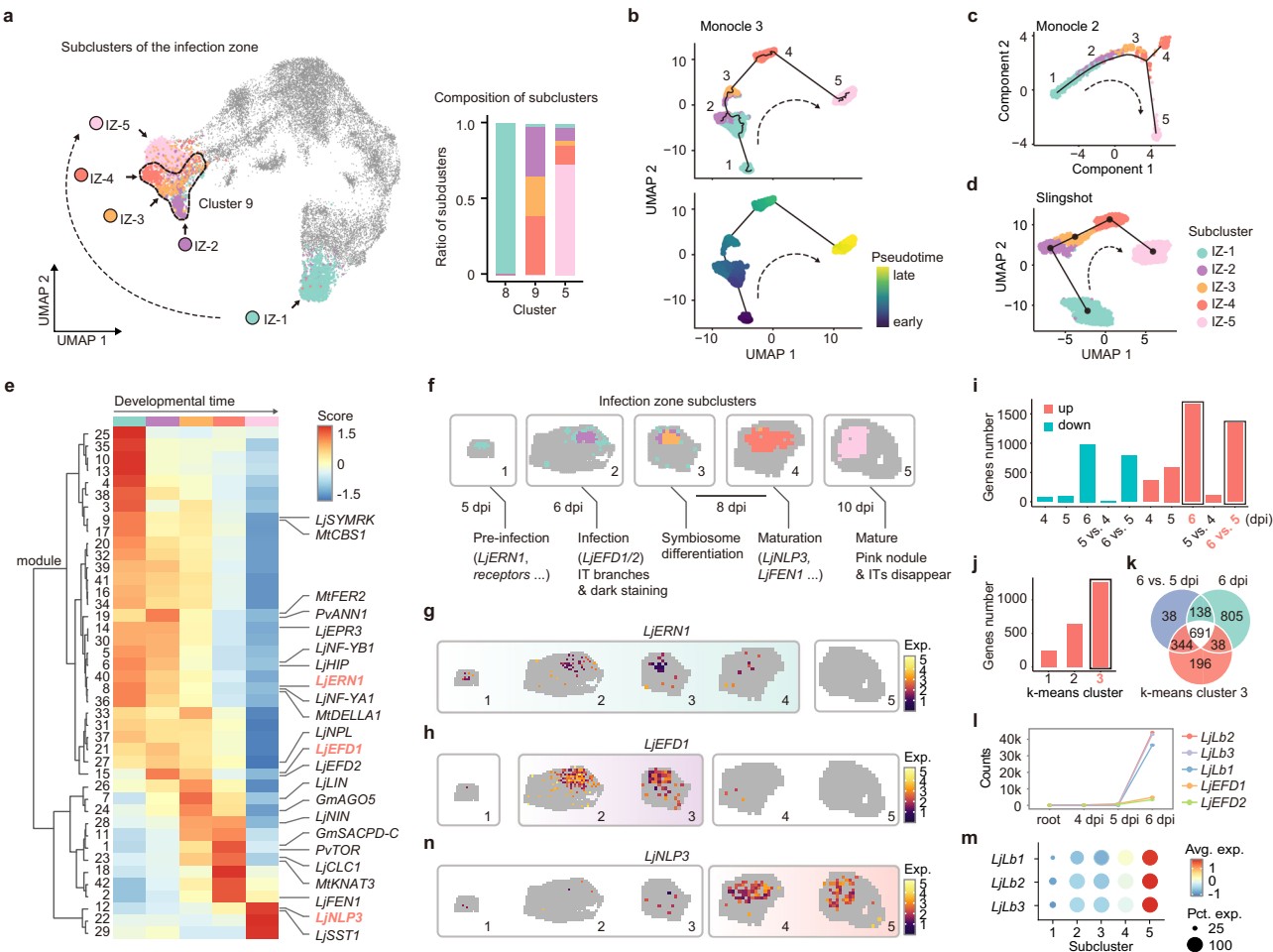

**Fig. 2 | Exploring infection zone development at the molecular level. a** The results of sub-clustering analysis on the infection zone (clusters 5, 8, and 9). The UMAP shows the distribution of subclusters within major clusters. Bins do not belong to the infection zone are labeled as gray. Cluster 9 is circled by dotted line, illustrating its division into three subclusters. The arrowed curve indicates the developmental direction. The bar plot shows the proportion of subclusters within clusters 5, 8 and 9. The marker gene list is provided in Supplementary Data 3. **b–d** The pseudotime trajectories of the infection zone. The pseudotime trajectories presented in panels (**b**), (**c**), and (**d**) are derived using the R packages Monocle 3, Monocle 2, and Slingshot, respectively. The black lines represent the trajectories, while the arrowed black dotted lines indicate the developmental direction. The bins are color-coded to reflect the subcluster membership. The pseudotime progression obtained from Monocle 3 is displayed as an example (**b**). **e** Co-regulated modules of differentially expressed genes. The subclusters are arranged according to the developmental time. Previously investigated genes, or their orthologs, are listed on the right; genes analyzed in this study are red-colored; orthologs not assessed in

this study are represented with their original names. The detailed gene list is provided in Supplementary Data 3. **f** The representative images of spatially visualized IZ subclusters. The main biological processes are annotated based on functional genes at each stage. **g**, **h**, **n** The representative images of spatially visualized *LjERN1*, *LjEFD1* and *LjNLP3*. Their main expression stages are indicated by a distinct color background. **i** Statistics of differentially expressed genes in bulk transcriptomic analyses. '4 dpi', '5 dpi' and '6 dpi' represent inoculated roots at indicated versus untreated roots. '5 vs. 4 dpi' and '6 vs. 5 dpi' represent comparison between indicated inoculated samples. The detailed gene list is provided in Supplementary Data 3. **j** Statistics of genes belonging to each k-means cluster in bulk transcriptomic analyses. The detailed gene list is provided in Supplementary Data 3. **k** Venn diagram showing the intersections of genes from indicated groups from **i** and **j**. **l** Line graph showing the expression of indicated genes. **m** Bubble plot showing the clustering expression pattern of *LjLb*s in the spatial transcriptomic data.

(cluster 8) and mature stages (cluster 5) remained almost unchanged, corresponding to subclusters IZ-1 and IZ-5, respectively. In contrast, cluster 9 was dissected into 3 subclusters, implying that differentiation, after infection, had multiple developmental stages. The pseudotime trajectories were next constructed for the infection zone, illustrating the developmental direction of these bins at the molecular level (Fig. 2b–d, and Supplementary Fig. 4a–c)[20]. Both the subclustering and trajectory results aligned with developmental time (Supplementary Fig. 4d). Furthermore, we selected representative GO terms for the infection zone from the above analyses (Supplementary Fig. 3c, e and f) and examined their relative enrichment in each subcluster (Supplementary Fig. 4e). These results also matched the developmental time. For example, subcluster IZ-1 corresponded to the earliest stage, and the GO terms 'nucleosome' and 'ribosome',

indicative of early developmental processes, were most prominently enriched within this subcluster (Supplementary Fig. 4e).

Differential expression analysis, based on the sub-clustering information, was next performed to identify genes that vary during development and genes with similar expression patterns were grouped into co-regulated modules (Fig. 2e). The investigated genes and orthologs were annotated to provide insights into associated biological processes during nodule development (Fig. 2f)[31]. For instance, *ERN1* (*ERF REQUIRED FOR NODULATION 1*), an important transcription factor (TF) regulating the progression of the infection thread (IT), began its expression at pre-infection stage, as previously reported[32–34], and ceased to be expressed at 10 dpi (Fig. 2g), consistent with our anatomical observation that IT branches were barely detectable during this stage (Supplementary Fig. 1d). Furthermore, MtNF-YA1

(NUCLEAR FACTOR Y SUBUNIT A1) and MtDELLA1 were found to cooperate in activating *MtERN1*[35]. We detected *LjNF-YA1* and the ortholog of *MtDELLA1* in the same module with *LjERN1* (Fig. 2e), suggesting the conservation of this regulation.

After the pre-infection stage, rhizobia invade into plant cells, and are enveloped by plant membrane[8,17,18]. This progression is in alignment with subcluster IZ-2 at 6 dpi (Supplementary Fig. 1c and 4d). Notably, several marker genes from this subcluster are associated with membrane trafficking, including calreticulin, which has been detected on the symbiosome membrane (Supplementary Fig. 4f)[36]. These genes could offer insights into understanding the mechanism of symbiosome formation and the endocytosis pathway involved. In the differential expression analysis, module 15 showed a specific peak during this stage and harbored two homologs of *MtEFD* (ERF REQUIRED FOR NODULE DIFFERENTIATION) (Fig. 2e), an important TF expressed in the nodule apical zone II and regulates differentiation processes in *M. truncatula*[7,37]. Spatial visualization confirmed that *LjEFDs* were induced during the infection stage (Fig. 2h), suggesting that EFDs regulate postinfection differentiation processes in *L. japonicus* nodules.

Considering that the infection event is crucial for symbiosis, and has been recognized as the pivotal event triggering transcriptional changes[38,39], we further conducted traditional bulk transcriptomic analyses by sampling the whole root, including nodules, at 4, 5, and 6 dpi (Supplementary Fig. 4g, h). PCA analysis revealed that the inoculation with rhizobia altered the transcriptome of roots, particularly after 6 days (Supplementary Fig. 4i). Differential expression analysis and k-means clustering analysis also indicated a significant change in the transcriptome at 6 dpi (Fig. 2i, j; Supplementary Fig. 4j–l). These findings confirmed our identification that infection begins at 6 dpi, and supported the notion that infection significantly alters the transcriptome.

Interestingly, *Lbs* were among the genes that were differentially expressed at 6 dpi (Fig. 2k, l). This contrasts with a previous study on *M. truncatula*, which reported that *Lbs* initiate expression before maturity[40]. Our spatial transcriptome analysis indicated that *Lbs* were induced at stage IZ-2 and significantly further upregulated at the mature stage (Fig. 2m). It is likely that the previous study only detected the high expression of *Lbs* at this stage. Furthermore, although the above membrane trafficking-related genes were induced at 6 dpi, most of them were not significantly differentially expressed, because they were also highly expressed in certain tissues of the root (Supplementary Fig. 4f, m). These findings underscore the advantages of spatial transcriptomics.

## LjNLP3 transitions nodules from the process of differentiation to maturation

Investigations on the regulatory mechanisms underlying the transition from the differentiation to maturation process are limited[2,38]. We proposed that genes involved most likely appeared in co-regulated modules with a late expression pattern, such as modules 2 and 12, which contained several genes required for mature nodules (Fig. 2e)[24,25,41]. Within module 12, we identified *LjNLP3*, a member belonging to one of the most important gene families, NIN-LIKE PROTEIN, in nodulation (Fig. 2n). Through *LjNLP3* promoter-driven GUS activity assays, we confirmed that *LjNLP3* was induced after 8 dpi, at the late developmental stages (Supplementary Fig. 5a).

A previous study indicated that the LjNLP3 ortholog in *M. truncatula*, MtNLP2, plays an important role in mature nodules by inducing the expression of *Lbs*[16]. Our transcription activity assays indicated that LjNLP3 also regulated the expression of *LjLbs* in mature nodules of *L. japonicus* (Supplementary Fig. 5b). Combining with its expression period, *LjNLP3* is responsible for the above-elaborated enhancement of *LjLbs* (Fig. 2m).

In *M. truncatula*, mutation of *MtNLP2* leads to reduced plant biomass, and develops nodules less pink[16]. Here, we discovered that mutation of *LjNLP3* also decreased plant size, and the maturity of

nodules was a bit less at 10 dpi (Supplementary Fig. 5c–i). To further assess whether *LjNLP3* regulated nodule entry into the maturation phase, we overexpressed it in *L. japonicus*. We proposed that, if *LjNLP3* possessed this ability, premature expression of *LjNLP3* would drive the rudimentary nodules to enter the maturation phase, leading to a reduction in both the number and size of mature nodules. Our results illustrated that, indeed, overexpressing *LjNLP3* reduced nodule number and size (Supplementary Fig. 5j–l). Importantly, these smaller nodules retained their pink color (Supplementary Fig. 5j), and tissue sectioning assays revealed their similarity to wild-type nodules (Supplementary Fig. 5m), confirming their maturity. These findings supported our hypothesis that LjNLP3 regulates nodule entry into the maturation phase.

To transition nodules from differentiation to maturation, LjNLP3 should repress differentiation-stage genes. Previous studies demonstrated that LjNLP4 interacts with LjNIN (NODULE INCEPTION), inhibiting its ability to induce positive regulators in nodulation, such as *LjNF-Ys*, thereby preventing nodule development when nitrogen nutrient is added[42]. We established that LjNLP3 also interacted with LjNIN (Supplementary Fig. 5n, o), and interfered with LjNIN-induced expression of *LjNF-Ys* at the late developmental stages (Supplementary Fig. 5p–s). These results confirmed that LjNLP3 is not only important for mature nodule[16], but also critical in orchestrating the transition of nodules from the differentiation to maturation phase (Supplementary Fig. 5t).

## SCR and PLT are promising candidates for regulating the progression of nodule meristems

In addition to the infection zone, peripheral tissues are also crucial for nodulation, facilitating material exchange between the legumes and rhizobia. However, investigations into their development are limited[2,17]. Thus, we combined clusters 4 (meristems) and 7 (peripheral tissues) to investigate the development from meristems to peripheral tissues. Here, we obtained three subclusters aligned with pseudotime and developmental time (Fig. 3a–c, and Supplementary Fig. 6a–e). The earliest, PT-1, closely resembled cluster 4, while PT-2 and PT-3 corresponded to differentiating and mature peripheral tissues, respectively (Supplementary Fig. 6b). Notably, subcluster PT-2 emerged at 6 dpi, resembling the emerging NVT (Supplementary Fig. 6e). This was consistent with our anatomical observation (Supplementary Fig. 1c), providing solid evidence for previous speculations that peripheral tissues differentiation coincided with NVT emergence[17,43].

We next performed a differential expression analysis, and discovered an SCR (SCARECROW) in the module with an early expression pattern (Supplementary Fig. 6f). It is also among the exclusive marker genes of cluster 4 (Supplementary Fig. 3b). Previous studies in *M. truncatula* indicated that *MtSCR* is induced by infection and governs nodule meristem formation[6,44]. *LjSCR1* promoter-driven GUS activity confirmed that *LjSCR1* was highly expressed in nodule meristems, from 5 to 6 dpi (Supplementary Fig. 6g), suggesting that SCR's role in regulating nodule meristems is likely conserved across legumes.

Furthermore, we discovered that a *PLT* (PLETHORA) was co-expressed with *LjSCR1*, in the early stages, and another family member, *LjPLT2*, was in an adjacent module with a similar expression pattern (Supplementary Fig. 6f). In *M. truncatula*, *PLTs* were shown to be co-expressed with *SCR* in the meristems, and participate in regulating meristem organization[6,45]. Furthermore, in Arabidopsis, *AtSCR* and *AtPLT* cooperate in regulating the progression of root stem cells[46]. Collectively, these findings supported the notion that SCR and PLT worked cooperatively to regulate the progression of meristems in root nodules.

## Peripheral tissues and the infection zone differentiate, coordinately, in a determinate nodule

When combining the trajectories of the infection zone and peripheral tissues, we observed that subcluster PT-2 emerged concurrently with

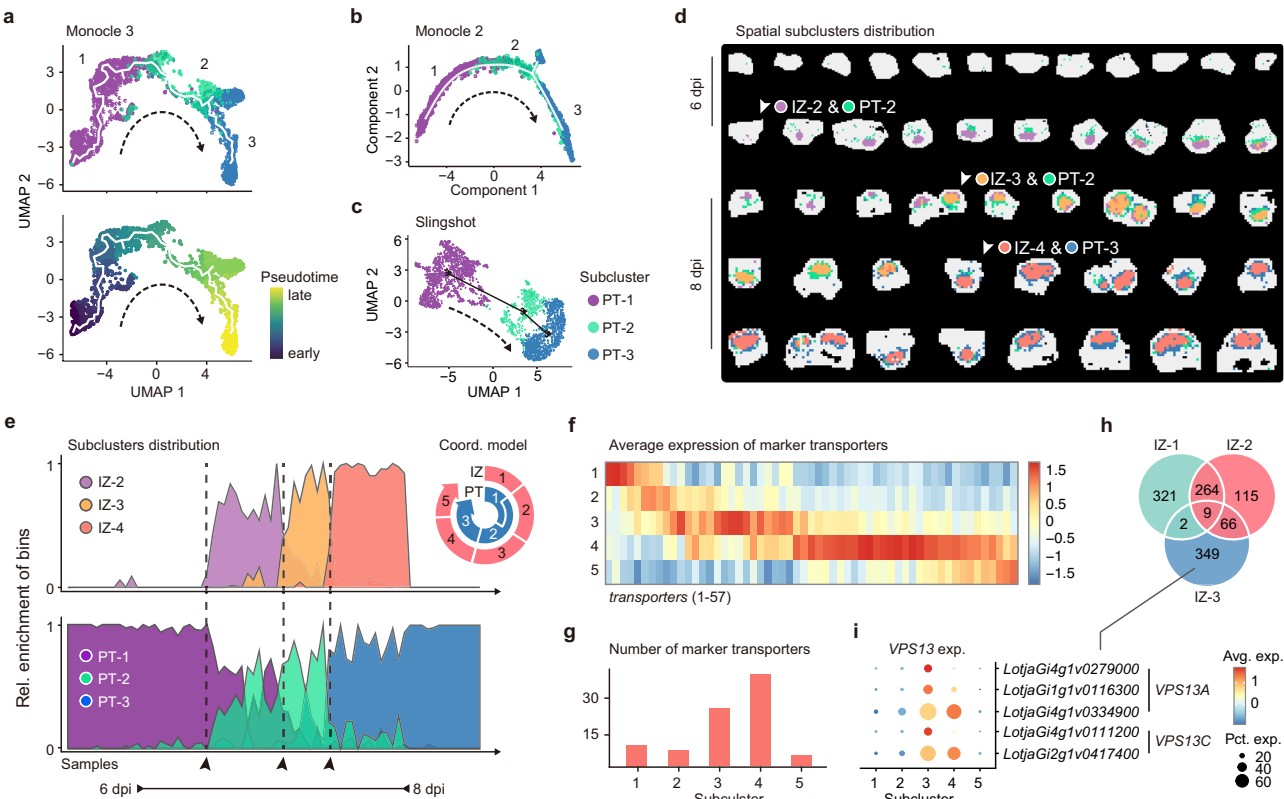

**Fig. 3 | Peripheral tissues and the infection zone differentiate, coordinately, in a determinate nodule. a–c** The visualization of pseudotime trajectories and related sub-clustering information of the peripheral tissues. The pseudotime trajectories presented in **a–c** are derived using the R packages Monocle 3, Monocle 2, and Slingshot, respectively. The solid white lines represent the trajectories, while the arrowed black dotted lines indicate the developmental direction. The bins are color-coded to reflect the subcluster membership. The pseudotime progression obtained from Monocle 3 is displayed as an example (**a**). **d** The spatial visualization demonstrates the coordinated differentiation of the peripheral tissues and infection zone. White arrowheads indicate transitional stages. **e** The area graph displays the coordinated trend of bins from PT and IZ subclusters. The calculation of relative enrichment per sample is described in the Methods section. The *x* axis represents all samples organized as shown in Fig. 1b, reflecting the developmental direction. Samples from 6 to 8 dpi are indicated by double-headed arrowheads, with transitional stages marked by black arrows, the same as in **d**. **f, g** The expression pattern of transporters in the infection zone. The heatmap (**f**) illustrates the clustering expression pattern of 57 annotated transporters in each subcluster. The bar plot (**g**) indicates the number of marker genes annotated as transporters in each IZ subcluster. Detailed information is provided in Supplementary Data 3. **h** Venn diagram showing the intersections of marker genes in subclusters 1, 2, and 3. **i** Bubble plot showing the clustering expression pattern of *VPS*s in the spatial transcriptomic data.

IZ-2 (Fig. 3d, e). These two subclusters represented the initial differentiation of peripheral tissues and the infection zone, respectively. Previous studies on soybean showed that NVT and infected cells likely appeared at the same stage, implying a simultaneous differentiation of these two regions[43]. Our findings provide molecular evidence for this synchronicity in determinate nodules.

The subsequent peripheral tissues subcluster, PT-3, represented the mature stage. In contrast, the infection zone had two additional developmental stages, IZ-3 and IZ-4, before reaching maturity (Fig. 3d, e). A comparison of marker genes in IZ subclusters revealed a significant increase in transporter expression in IZ-3 and IZ-4, with the highest levels observed in IZ-4 (Fig. 3f, g). Notably, the maturation of symbiosomes involves the activation of numerous transporters located on the symbiosome membrane, facilitating material exchange between legumes and rhizobia[24]. Our findings identified the IZ-3 stage as the likely initiation phase for this process, with several *VPS*s (*VACUOLAR PROTEIN SORTING*) emerging as distinct marker genes to the IZ-3 subcluster (Fig. 3h, i). All these *VPS*s belonged to the VPS13 family, which are important for membrane traffic of vacuole[47,48], and thereby, shed light on the mechanism underlying symbiosome maturation. Furthermore, the emergence of subcluster IZ-4 coincided with PT-3, aligning with the full development of nodule vascular bundles (Fig. 3d, e), indicating that the

heightened material exchange demand depended on fully functional peripheral tissues.

Previous research argued that nitrogen fixation would commence only after the NVT was competent to transport assimilates[17,49,50]. Our findings provided molecular evidence in support of this speculation, demonstrating that the maturation process of the infection zone, namely the IZ-4 stage, commenced subsequent to the establishment of NVT. Furthermore, our findings illustrated that the progressive symbiotic material demand throughout the entire nodule development, not solely confined to the maturity stage, was exquisitely met through the coordinated differentiation of the infection zone and peripheral tissues.

### Conserved regulatory genes imply evolutionary continuity and divergence between *L. japonicus* and *M. truncatula*

After clarifying the developmental trajectory of the *L. japonicus* nodule and the functional genes involved, we set our sights on the monophyly of Fabaceae, which is a fundamental theory for developing novel nitrogen-fixing crops by uncovering "lost" genes[14,15]. Sufficient genomic research has supported this theory. However, the complexity arose from the fact that legumes contain determinate and indeterminate nodules, which are morphologically different, posing a challenge in discerning their conservation from the perspective of developmental biology.

Previous research on *M. truncatula* categorized the root nodule into several zones, which correspond to different developmental stages. Significantly, several well-studied regulatory genes, as elaborated above, can characterize these distinct zones[5–7,51]. Here, we discovered that the orthologs of these crucial genes also exhibited specific expression patterns within *L. japonicus* (Supplementary Fig. 7a, b). This observation strongly suggested that these two species shared analogous developmental trajectories, despite their different nodule types. Thus, our findings contributed to a comprehensive understanding of the monophyletic theory from the perspective of developmental biology. Moreover, we have defined the biological processes during various developmental stages, which can help us understand the development of other types of nodules and discover new regulatory genes. For example, there have been no reports of TFs that regulate the initiation of symbiosome development at the IZ-3 stage. By annotating marker genes in IZ-3 subcluster, we discovered multiple candidate TFs likely involved in regulating this process (Supplementary Fig. 7c, d).

The most prominent divergence between determinate and indeterminate nodules lies in the presence of continuous meristems. For determinate nodules, inhibitory mechanisms should exist acting on meristemic activity. To pinpoint these regulatory genes, we examined genes initiating expression in subcluster PT-2, corresponding to differentiation initiation (Supplementary Fig. 6f). One gene of particular interest was a homolog of Arabidopsis, BIB (BALDIBIS) (Supplementary Fig. 6h–j). Previous studies indicated that AtBIB and its paralog, AtJKD, participate in determining the fate of root ground tissue[52], with one important role being the inhibition of cell divisions induced by AtSCR[53]. Based on these findings, we speculated that LjsymBIB also inhibited stemness and promoted nodule differentiation. It is noteworthy that our bulk transcriptomes showed that, unlike its ortholog in Arabidopsis, *LjsymBIB* was not expressed in root tissues, and has been specialized as a nodule regulatory gene (Supplementary Fig. 6k, i).

### Genes involved in nodule organogenesis revealed by spatial co-expression module analysis

Following our trajectory-based study on specific tissues, we conducted spatial co-expression module analysis to explore the transcriptional network of nodule organogenesis, from a global spatial perspective[54]. Twelve modules were obtained (Fig. 4a). The overall expression pattern of genes, within each module, was reflected by the spatial-visualized metagene enrichment score (Supplementary Fig. 8a). Our primary interest was in the identification of novel functional genes that might not have been revealed through unsupervised clustering analyses.

Among these candidates, genes within module S3 stood out, displaying unique expression patterns primarily observed in the early root cortex surrounding each nodule (Supplementary Fig. 8a). Within them, we identified four investigated nodulation-related (iso)flavonoid enzymes (Fig. 4a). Previous studies had shown that these genes are expressed in the root cortex, to produce flavonoids to attract rhizobia under low-nitrogen conditions, and are crucial for the initiation of nodulation[2,55–57]. However, information on TFs involved in regulating flavonoid metabolism was lacking. Thus, we analyzed TFs within module S3 and identified several WRKYs that displayed similar expression patterns (Fig. 4b and Supplementary Fig. 8b), consistent with earlier studies showing that WRKYs participate in flavonoid metabolism[58]. Therefore, we considered these WRKYs to be promising candidates that redundantly regulated flavonoid metabolism during nodulation.

Module S1 contained the well-known NIN (Fig. 4a), a critical TF for nodulation[59]. We hypothesized that TFs co-expressed with *NIN* might also play important roles in nodulation. Thus, we annotated TFs in module S1 and identified a *LHW* (LONESOME HIGHWAY) with the highest correlation score (Fig. 4c). Spatial visualization and *LjsymLHW* promoter-driven GUS activity assays confirmed that its expression

pattern was similar to that of *LjNIN* (Fig. 4d, e). Previous research indicated that neofunctionalization of genes participating in root development as one important strategy for legumes to achieve nodule organogenesis[2,14,15,38]. LHW is a key TF that regulates root development[60–62], which makes *LjsymLHW* a promising candidate in regulating nodule development. It is noteworthy that there were two LHW homologs in *L. japonicus* (Fig. 4f and Supplementary Fig. 8c). Similar to *LjsymBIB*, *LjsymLHW* is primarily expressed in nodules (Fig. 4g, h, and Supplementary Fig. 8d). This suggests that *LjsymLHW* has also evolved as a nodule regulatory gene. Comparative analysis of homologs, focusing on features such as amino acid sequence (Supplementary Fig. 9), can offer valuable insights into the neofunctionalization of genes involved in nodulation.

## Discussion

SNF is an intricate process in which legumes cooperate with rhizobia to overcome nitrogen shortage. Harnessing the power of SNF not only promises to reduce our reliance on fossil fuels but also offers a beacon of environmental responsibility by mitigating pollution. Understanding the mechanism of nodule organogenesis is the foundation for accomplishing this purpose[1,2]. Here, by using the cutting-edge spatial transcriptomics, we included 72 nodule samples to construct a comprehensive transcriptomic atlas of *L. japonicus* nodule (Supplementary Fig. 10).

Our study has traced the developmental trajectories of the infection zone and peripheral tissues, at the molecular level, revealing previously unobservable details. A considerable number of regulatory genes for various developmental stages were identified. Of particular interest are genes regulating the infection zone, which is the central site for symbiosis, and our molecular evidence indicated that it stands out as being distinct (Supplementary Fig. 3a). As a result, the discovery and characterization of functional genes in the infection zone offer exciting prospects for unveiling the intricacies of this symbiotic relationship. Significantly, we uncovered such genes and offered detailed information on their respective expression stages, each linked to specific biological processes, which would facilitate functional investigations on them, as exemplified by our study on *LjNLP3*.

In addition to symbiosis, material exchange is the other central issue for SNF, and knowledge of how the legume and rhizobium achieve exquisite integration of metabolism is critical for understanding nodulation[25,63]. In our study, we identified three main post-infection developmental stages within the infection zone, which can be characterized by progressive expression of transporters (Fig. 3f). Coincidently, there are several main developmental phases for the rhizobium during its interaction with legumes: the invasion, duplication, and differentiation stages[5,8,63]. This progressive expression of transporters reflects the dynamic symbiotic material demands, and functional studies on these transporters can unveil the synchronization of metabolism between legumes and rhizobium. Meanwhile, as rhizobium is the other member in this symbiosis, investigation from the bacteria perspective can further provide insights. The development of spatial transcriptomic techniques that could capture prokaryotic mRNA would offer a platform for such studies.

In summary, our work delineates the developmental process of the determinate nodule, shedding light on the poorly understood molecular events during nodule organogenesis. We believe that this comprehensive transcriptomic atlas will accelerate research in this field and thereby facilitate agricultural applications.

## Methods

### Plant growth and nodulation conditions

We used the *L. japonicus* ecotypes Gifu B-129 for this study. To prepare Stereo-seq samples, seeds were scarified and then surface-sterilized with NaClO for 10 min, followed by four washes with sterile water. These seeds were then placed on filter paper and kept moist for two

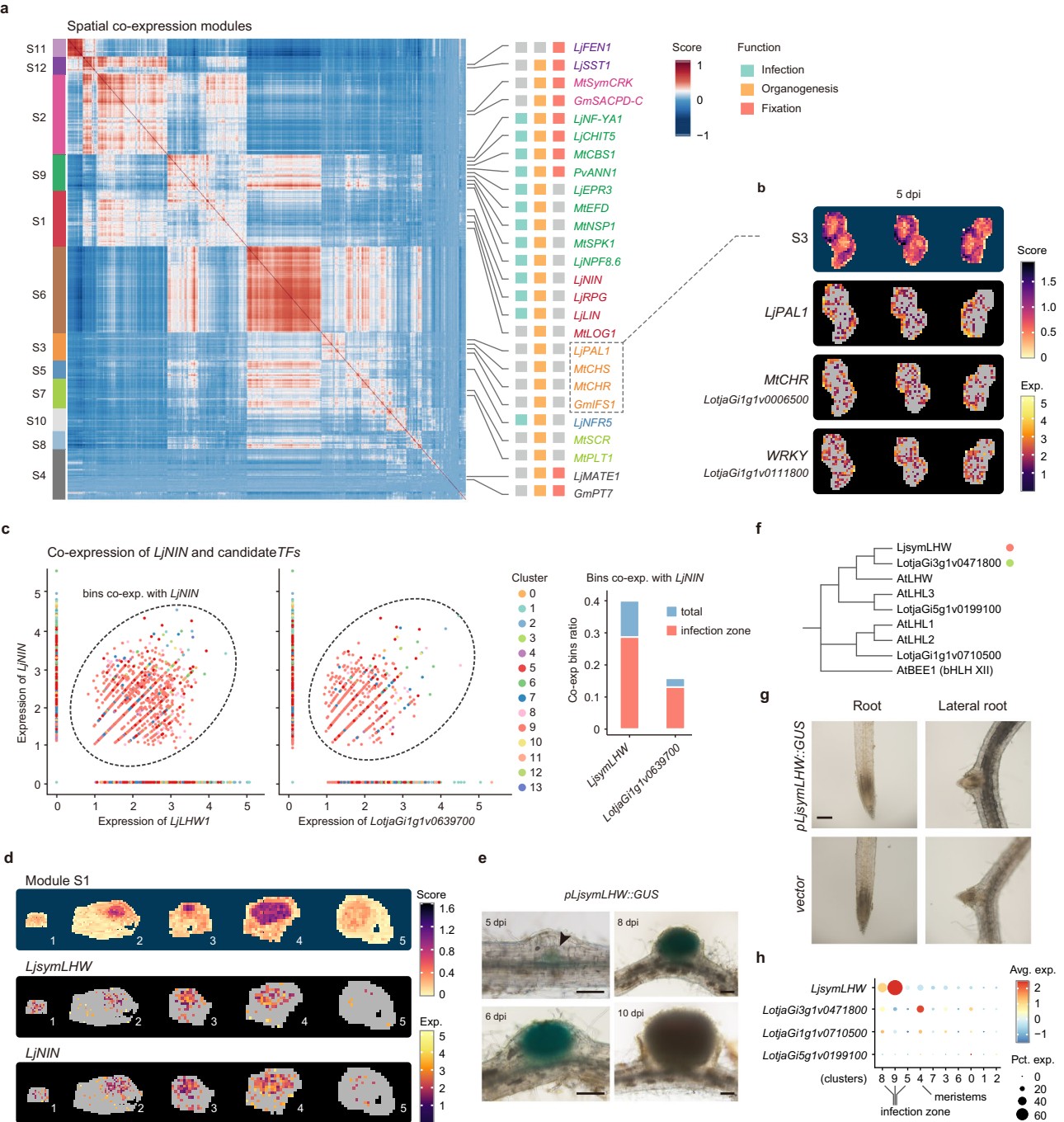

**Fig. 4 | Genes involved in nodule organogenesis revealed by spatial co-expression module analysis. a** Heatmap displaying the spatial co-expression modules of candidate genes involved in nodulation. Investigated genes or orthologs participating in SNF are listed alongside the corresponding modules, with their main function concisely represented by infection, organogenesis, and fixation. The full gene list is available in Supplementary Data 5. **b** Spatial visualization of representative genes in module S3 and corresponding spatial metagene enrichment scores.**c** Scatter plot displaying the correlated expression of *LjsymLHW* and *LjNIN* in the infection zone. Bins are colored based on clusters. Bins that simultaneously express *LjNIN* and *LjsymLHW* are circled; the total count of these bins, along with the count of bins that belong to the infection zone clusters, is presented. The second highest correlation gene *LotjaGi1g1v0639700* is displayed as control. The

full list of co-expressed TFs is provided in Supplementary Data 5. **d** Representative images of spatial visualization of *LjNIN* and *LjsymLHW*. As these two genes are expressed in the infection zone, we selected the representative samples demonstrated in Fig. 2f. **e** GUS activity showing the expression of *LjsymLHW* in root nodules. Hairy roots transformed with *pLjsymLHW::GUS* were imaged at the indicated dpi. The GUS signal in the nodule primordium is indicated by arrowheads at 5 dpi. Experiments were repeated three times with similar results. Scale bar = 200 μm. **f** Phylogenetic tree of genes belonging to LHW subfamily from *L. japonicus* and Arabidopsis. **g** GUS activity showing that *LjsymLHW* is barely detected in roots. Experiments were repeated three times with similar results. Scale bar = 200 μm. **h** Bubble plot showing the clustering expression pattern of *LjsymLHW* and its paralogs in the spatial transcriptomic data.

days at 4 °C, then fully-imbibed seeds were planted in autoclaved vermiculite and perlite (2:1) mixed with Nitrogen-free nutrient solution[64], and grown in a growth chamber under a 16-h light/8-h dark cycle, at 24 °C. After 10 days, the seedlings were inoculated with

rhizobium strain *Mesorhizobium loti* MAFF303099 carrying GFP fluorescence. The strains were grown for 2 days at 28 °C on solid TY plates, containing 5 mg mL⁻¹) tetracycline, and then resuspended in Nitrogen-free nutrient solution with $OD_{600}$ diluted to 0.01.

For hairy root transformation, positive seedlings were grown for 5 days and then inoculated with *M. loti* MAFF303099. For nitrate treatment conditions, 15-day-old seedlings were inoculated with *M. loti* MAFF303099 or 5 mM nitrate. And then watered corresponding nutrient solution with or without nitrate every 4 days. At 5 dpi and 10 dpi, whole roots were sampled.

## Plasmids construction

For hair root transformation, plasmid skeleton pKGW-MCS was constructed by adding multiple cloning sites (*Afe*I, *Sal*I, *Spe*I) into the pKGW-RR-MGW vector containing the DsRed selection marker[65]. The sequence was then PCR amplified from this pKGW-RR-MGW vector, using primers *pKGW-mcs-1F*, *pKGW-mcs-1R*, and then cloned through an In-Fusion® HD Cloning Kit (Clontech) into a pKGW-RR-MGW vector digested by *Pme*I and *Aat*II. Two fragment sequences (*attR1-CmR-ccdB-attR2* and *GUS-NOS*) were amplified, using primers *pKGW-GUS-1F* and *pKGW-GUS-1R*, *pKGW-GUS-2F* and *pKGW-GUS-2R* from pEarleyGate-103 and PBI-121, respectively. The amplified sequences and pKGW-MCS digested by *Sal*I and *Spe*I were linked, using an In-Fusion® HD Cloning Kit. This destination vector was termed pKGW-ccdB-GUS. The elements of the Cauliflower mosaic virus *35 S* promoter (CaMV-35S), *attR1-CmR-ccdB-attR2*, and NOS terminator were PCR amplified, by primers pKGW-35S-ccdB-1F and *pKGW-35S-ccdB-1R*, *pKGW-35S-ccdB-2F* and *pKGW-35S-ccdB-2R*, *pKGW-35S-ccdB-3F* and *pKGW-35S-ccdB-3R*, and then cloned using an In-Fusion® HD Cloning Kit (Clontech) into a pKGW-MCS vector digested by *Sal*I and *Spe*I. This destination vector was termed pKGW-35S-ccdB.

For GUS staining assays, the *LjNLP3*, *LjSCR1*, *LjsymBIB* and *LjsymLHW* promoters were amplified, using primers *pNLP3-DONR-F* and *pNLP3-DONR-R*, *pSCR-DONR-F* and *pSCR-DONR-R*, *pLHW-DONR-F* and *pLHW-DONR-R*, respectively. These promoters were then cloned into pDONR221, by BP reaction (Invitrogen), to generate entry vectors that were then cloned into pKGW-ccdB-GUS, by LR reaction (Invitrogen). For hairy root transformation using the *LjNLP3* overexpression construct, a full-length CDS of *LjNLP3* was amplified, using *NLP3-OE-DONR-F* and *NLP3-OE-DONR-R* primers, and then cloned into pKGW-35S-ccdB through BP and LR reactions.

To construct plasmids for Yeast Two-Hybrid assays, the full-length CDS of *LjNIN*, *LjNLP3* and the PB1 domain of *LjNIN*, were amplified and cloned into pGBKT7 and/or pGADT7 (Clontech), using the In-Fusion® HD Cloning Kit. The pCAMBIA1300-nLUC and pCAMBIA1300-cLUC vectors were used in split-luciferase complementation assays. The *LjNLP3* and *LjNIN* were amplified, using primers *LjNLP3-nLUC-F* and *LjNLP3-nLUC-R*, *LjNIN-cLUC-F* and *LjNIN-cLUC-R*, and then cloned into the corresponding skeleton vectors digested with *Kpn*I and *Sal*I, using an In-Fusion® HD Cloning Kit.

To generate a skeleton plasmid PUC57-transactiveQuant backbone, containing the CaMV-*35S* promoter, coding regions of Renilla luciferase and Firefly luciferase, NOS terminator, the corresponding elements were synthesized and assembled by Sangon Biotech (Shanghai) Co., Ltd. (Supplementary information, Supplementary Data 6). The *LjNIR1*, *LjNF-YB1* and leghemoglobins (*LjLb2* and *LjLb3*) promoters were amplified using relevant primers and cloned into PUC57-transactiveQuant by In-Fusion® HD Cloning Kit through the *Spe*I and *Hap*I double digestion sites. Plasmids of pBeacon-HA and pBeacon-FLAG were used to transiently express indicated proteins in Arabidopsis protoplast[66]. *LjNLP3* and *LjNIN* were cloned into pDONR221, by the BP reaction, and then into pBeacon-HA and pBeacon-FLAG by LR reaction, respectively.

The sequences for all primers used in this study are listed in Supplementary Information, Supplementary Data 6.

## Yeast two-hybrid assays

The Y2H assays were conducted according to previously described methods[67]. In brief, pGADT7-LjLHW and pGBKT7-LjNIN-PB1 were co-transformed into yeast strain AH109 (Clontech), or transformed separately with the complementary empty vector. The yeast transformants were screened on SD (Synthetic Dropout) media lacking leucine and tryptophan (SD/-Leu-Trp). Protein interaction and self-activation were assessed by screening yeast transformants on SD media lacking leucine, tryptophan, histidine and adenosine (SD/-Leu-Trp-His-Ade).

## Split-luciferase complementation assays

The EHA105 strains carrying each of the two constructs, pCAMBIA1300-cLUC-LjNIN and pCAMBIA1300-LjNLP3-nLUC, pCAMBIA1300-cLUC-LjNIN and pCAMBIA1300-nLUC, pCAMBIA1300-cLUC and pCAMBIA1300-LjNLP3-nLUC were co-transformed into *Nicotiana benthamiana* leaves, via agroinfiltration. Two days after infiltration, 1 mM luciferin (Promega) was sprayed onto leaves and images were captured using a CCD (TANON 5200) chemiluminescent imaging system.

## Dual-luciferase reporter assays

Protoplasts prepared from Arabidopsis leaves were used for dual-luciferase reporter assays. The leaves of 3-week-old plant were used to isolate protoplasts. Pre-preparation enzyme solution containing 20 mM MES (pH 5.7), 1.5% cellulase RS, 0.4% macerozyme R-10, 0.4 M mannitol and 20 mM KCl and warm at 55°C for 10 min. After cooled solution, add 10 mM $CaCl_2$, 2.5 mM β-mercaptoethanol and 0.1% BSA. The leaves were cut into 1–2 mm strips, and then incubated into enzyme solution and vacuum-inflitrated with 600 mbar for 20 min under a desiccator (Thermo, VT6025). The strips continue digestion on a shaker at 40 rpm under room temperature for 3 h. Protoplasts released from enzyme solution were collected by centrifugation with 150 g (Eppendorf, 5810 R) and transfected with indicated vectors via the polyethleneglycol-mediated procedure[68]. After incubation at room temperature for 12 h, the Renilla and Firefly luciferase activities were measured using the Dual-Luciferase Reporter Assay System (Promega). The ratio of Firefly luciferase to Renilla luciferase activity, FF/REN represented the transactivation extent of the protein of interest.

## Hairy root transformation of *L. japonicus*

Respective constructs, based on the pKGW-MCS skeleton vector containing DsRed, were introduced into *Agrobacterium rhizogenes* AR1193 strain. Hairy root transformation of *L. japonicus* was conducted, as previously described[69]. Positive plants were screened through resistance and detection of DsRed fluorescence, at the excitation wavelength 587 nm, under a fluorescent stereo microscope (Leica, M165).

## Histochemical GUS assays

The hairy root transformed plants carrying *pLjNLP3::GUS*, *pLjLHW::GUS*, or *pLjSCR::GUS*, were used for GUS staining assay. After inoculation, with or without *M. loti* MAFF303099, roots were excised, at the indicated times, and put into staining buffer and vacuum-inflitrated for 20 min, and then incubated for 2–3 h at 37 °C. After staining, images were taken on a stereo microscope (Leica, M165) and inverted fluorescence microscope (Olympus, IX83).

## Resin embedment and sectioning

At 5 dpi, 6 dpi, 8 dpi, and 10 dpi, nodules representing specific developmental stages were sampled from *L. japonicus* seedlings inoculated with *M. loti* MAFF303099 carrying the GFP reporter. Tissues were fixed, in glutaraldehyde/formaldehyde/50 mM sodium phosphate pH7.2 buffer (2:5:43, v/v/v), and then dehydrated in an ethanol gradient series, followed by embedment in Technovit 7100 resin[70]. Sections (7 mm) were cut using a rotary microtome (RWD, S700), and images were captured using an inverted fluorescence microscope (Olympus, IX83).

### *nlp3* mutant and genotyping

The *LORE1* insertion lines of *LjNLP3*, *nlp3* (plant ID is 30072312) was identified by reverse screening of the Tnt1-mutant collection from the *Lotus* Base (https://lotus.au.dk). The DNA was extracted from leaves according to routine CTAB method and used for genotyping with a *LjNLP3* specific reverse primer and *LORE1* forward primer (*P2* primer, which is provided on the *Lotus* Base website). To further confirm the effect of Tnt1 insertion on the function of LjNLP3, we designed amplification primers on flanking sequence of the insertion site, and the cDNA extracted from wild-type and *nlp3* mutant was used as template. All primers used for genotyping are listed in Supplementary Data 6.

### qRT-PCR analysis

Total RNA was extracted from whole roots, using TRIzol Reagent (Invitrogen), according to the manufacturer's instructions. The first-strand cDNA was synthesized by ReverTra Ace® qPCR RT Master Mix with gDNA Remover (TOYOBO). qRT-PCR analysis was performed using gene-specific primers (Supplementary information, Supplementary Data 6) on a QuantStudio 3 Applied Biosystems (Thermo Fisher Scientific), according to the manufacturer's instructions, using a 10 μL reaction system, including 1 μL diluted cDNA, 0.3 μM primers, and 5 μL ChamQ™ Universal SYBR qPCR Master Mix (Vazyme). The *LjUBQ* was used as a reference gene.

### Stereo-seq tissue sectioning and library building

Nodules from specific developmental stages were sampled and embedded, immediately, in Tissue-Tek® O.C.T Compound (optimal cutting temperature compound, SAKURA). All samples were placed on the bottom of the cassette so that they remained on the same plane. After bubble extraction, the samples were placed on dry ice to freeze. The frozen block was balanced at −20 °C for 30 min before being sectioned in a Cryostat Microtome. The slice thickness was set to 10 μm. A stereo-seq transcriptomics reagent set (BGI, 111ST114) was used and the manufacturer's instructions were followed with little modification[19]. In brief, methanol was used to fix the tissue on the chip for 30 minutes, the chip was stained with the assay reagent from the Qubit ssDNA Assay Kit (Invitrogen) and a photograph was taken to get the tissue position. The tissue on the chip was then permeabilized by 0.1% PR enzyme in the reagent set in 0.01 M HCl buffer (pH = 2) for 12 minutes at 37 °C, so that RNAs in the tissue could be captured by polyT. Reverse transcription was performed overnight in a 37 °C incubator, and the cDNA was released in the cDNA release buffer for 3 hours after removal of the tissue. The buffer was then collected for sequencing library construction. Stereo-seq library was prepared using a kit (BGI, 111KL114) and sequenced using a DNBSEQ-T5 sequencer.

### Stereo-seq raw data processing

The reference genome is *L. japonicus* (Gifu v1.3)[11,19,71]. Raw sequencing reads were mapped to genomes using the SAW software suite (https://github.com/BGIResearch/SAW) with default parameters. Reads that could be uniquely mapped to coding regions of the genome were counted. Every read contained a coordinate identity (CID) so the coordinate information was obtained. The ssDNA photo described above was used in the SAW workflow and the raw reads could be mapped back to their actual position on the tissue. Finally, a matrix of the expression count of every gene expressed in every spot was generated. The original spots were 220 nm diameter circles and the distance between every two spots is 500 nm. The resulting expression profile matrix files were processed into single-cell data using the Bin 20 method[11,19]. For tissues collected at 5 and 6 dpi, data from $50 \times 50$ spots area was combined into one bin (equivalent to $36 \times 36 \, \mu m^2$ area), which could cover one cell observed in our tissue sectioning analyses. While, a bin size of $80 \times 80$ (equivalent to $58 \times 58 \, \mu m^2$ area) was used for tissues collected at 8 and 10 dpi, as the size of infected cells expanded

at these stages (Supplementary Data 1). Finally, these bins were used as raw gene count matrices.

### Unsupervised clustering analysis

The raw gene count matrices for each sample were combined into a Seurat project using the merge() function. Unsupervised clustering analysis was started with quality control measures to filter out bins with a total number of UMI<50 or genes<100. The remaining bins were then processed using R package Seurat (v4.2.1)[72], including normalization, scaling, feature gene selection, PCA dimension reduction (top 30 PCs were used in the subsequent analyses), harmony dimension reduction, and clustering with a resolution parameter set to 0.6. For sub-clustering analyses on the infected zone and peripheral tissues, the resolution parameter was set to 0.25 and 0.1, respectively.

### Construction of clustering trees

The Seurat clustering results were used to construct a clustering tree, using the R package clustree (v0.5.0)[73], with resolutions ranging from 0 to 1.2 in increments of 0.2.

### Correlation analysis between Seurat clusters

The Seurat object was used to extract the top 2000 genes having the highest standard deviation and expression of these genes, within each Seurat cluster, was averaged to generate an expression matrix. Pearson's correlation coefficient (PCC) was then calculated between clusters using the expression matrix.

### GO enrichment analysis

The GO enrichment analysis was performed on the *Lotus* Base, using the *L. japonicus* genome (Gifu v1.3) annotation as the background[71].

### Pseudotime trajectory analysis

Pseudotime trajectory analysis was performed using the R package Monocle 3 (v1.2.9), Monocle 2 (v2.22.0), and Slingshot (2.12.0)[20,74,75].

In Monocle 3, we extracted Seurat objects corresponding to the infection zone or peripheral tissues and transformed them into Monocle objects, using the R package SeuratWrappers with the as.cell_data_set function. In line with the unsupervised clustering analysis conducted by Seurat, the top 30 primary components were utilized in the subsequent analyses (num_dim = 30). After pre-processing, dimensionality reduction (reduction_method = "UMAP"), and clustering, we constructed trajectories using the learnGraph() function.

In monocle 2, the gene-expression matrix was constructed with Seurat object by utilizing as.matrix function. Differentially expressed genes (DEGs) were identified with the differentialGeneTest function. The top 2000 genes with the lowest *q* value were used to construct the pseudotime trajectory.

In Slingshot, we utilized the as.SingleCellExperiment function to convert Seurat objects into SingleCellExperiment objects. Subsequently, cell lineages were defined using the slingshot function. The determined trajectories were projected onto UMAP obtained from Seurat for visualization.

### RNA-seq analysis

After inoculation, with or without *M. loti* MAFF303099, roots were sampled at the indicated time. Eight seedlings were pooled as one sample, and three biological replicates were used for RNA extraction and RNA sequencing. RNA-seq was performed on DNBSEQ-T7 platform (MGI).

Sequence quality of RNA-seq libraries was assessed using FastQC (v0.11.9). The adapter sequences and low-quality reads were filtered using fastp (v0.20.1)[76], with a quality threshold (-q) set at 20 and a length threshold (-u) at 30 to discard reads shorter than 30 bp. The clean reads were mapped to the *L. japonicus* (Gifu v1.3) reference genome using hisat2 (v2.1.0)[77], and the aligned reads were sorted using SAMtools

(v1.11)[78]. Gene-expression levels were quantified using featureCounts (v2.0.3)[79]. DEGs were evaluated using the R package DESeq2 (v1.38.3)[80] with adjustedP value<0.05 and $|\log_2(FoldChange)| \geq 1$. Finally, to discern trends in DEGs across samples, gene-expression patterns were determined through k-means clustering using all DEGs.

## Calculation on the relative enrichment of bins belonging to different subclusters in each sample

As an example, we calculated the relative enrichment of bins belonging to subcluster PT-1 using the following procedure. For sample $i$, the number of bins belonging to subcluster PT-1 is $count_i$ and the total number of bins belonging to this cluster across all samples is $\sum_{i=1}^{72} count_i$. The total number of bins belonging to the infection zone and peripheral tissues, in sample $i$, is $sum\_bin_i$, and the total number of bins belonging to these two tissues, across all samples, is $\sum_{i=1}^{72} sum\_bin_i$. The enrichment of bins belonging to subcluster PT-1, in sample $i$, is denoted as $E_i$ and calculated using the following formula:

$$E_i = \frac{count_i / \sum_{i=1}^{72} count_i}{(sum\_bin_i / \sum_{i=1}^{72} sum\_bin_i)} \qquad (1)$$

The relative enrichment was calculated by taking the sample with the highest value, as the denominator, and calculating the relative values of other samples.

## Spatial co-expression modules analysis

Spatial co-expression modules analysis was performed using the R package Giotto (v3.1)[54]. Firstly, we converted Seurat objects to Giotto objects using the seuratToGiotto function. Next, we detected spatially coherent expression patterns, using the binSpect function. Finally, we clustered the top 1000 genes into spatially co-expressed feature modules, using the clusterSpatialCorFeats function, with parameter $k = 12$.

## Annotation of investigated genes in the pseudotime trajectory analysis and spatial co-expression modules analysis

The protein sequences of investigated genes participating in root nodulation were carefully collected and blasted against their homologs, with the highest alignment score, in *L. japonicus* ecotype Gifu. Based on this list, we annotated the investigated genes.

## Reporting summary

Further information on research design is available in the Nature Portfolio Reporting Summary linked to this article.

# Data availability

The raw and processed data of Stereo-seq have been deposited to CNGB Nucleotide Sequence Archive (accession code: CNP0004336) and CNGB Spatial Transcript Omics DataBase (accession code: STT0000041). The raw bulk RNA-Seq data have been uploaded to NCBI, and can be accessed through the bioproject ID PRJNA1075833. Source data are provided with this paper.

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

## Acknowledgements

We thank Erwei Zuo (AGIS), Weicai Yang (UCAS) and Rene Geurts (WUR) for project discussion and editing assistance. We also acknowledge the assistance of the China National GeneBank (CNGB) and the STOmics Cloud. This work was supported by the National Key Research and Development Program of China under award number 2019YFA0906200 (S.H.), National Natural Science Foundation of China under award number 32070250 (Y.Z.), Shenzhen Outstanding Talents Training Fund (S.H.), and the Special Funds for Science Technology Innovation and Industrial Development of Shenzhen Dapeng New District Grand No. RC201901-05 (S.H.).

## Author contributions

J.Z., H.L., K.Y., and S.H. conceived and designed the project. K.Y., J.Z. and W.L. wrote the manuscript. K.Y., L.Z., Z.D. and Z.M. performed the bioinformatics analyses. J.Z., F.B., L.Z., and Z.T. carried out the molecular and biochemical experiments. Y.Z., X.Y., X.X., and E.W. coordinated the project.

## Competing interests

The authors declare no competing interest.
