## [Peer Review File · Nature Communications]

REVIEWER COMMENTS

Reviewer #2 (Remarks to the Author):

This reviewer wants to thank the authors for answering the comments and for sharing Supplemental Table 1. This reviewer understands that the authors analyzed 172 million of unique reads (less than 1.5% of the 11.69 giga sequenced reads which is surprising to this reviewer) across 19,872 bins. This reviewer needs more information to better assess the quality of the data including the average numbers of expressed genes and UMI per bin.

Besides, this reviewer does not understand the rationale for conducting a bulk transcriptomic experiment in response to reviewer #2's first comment. The authors seem to agree that the used spatial transcriptomics is not a technology that supports single-cell resolution transcriptomic analyses. Considering that the present study generated 19,872 bins across 72 cross-sections, it means that each cross-section is composed of an average of 267 bins of 36x36 μm^2 to 60x60 μm^2 . This reviewer assumes that many bins of such area cannot cover one cell as suggested by the authors (lines 611-612).

The lack of resolution might affect the interpretation of the data. For instance, this reviewer is not certain that this study is truly "an accurate, tissue-resolution molecular map" as claimed by the authors because each bin might not reflect a group of cells sharing similar functions [e.g., this reviewer did not notice the group of the uninfected cells of the infection zone of the *L. japonicus* nodule or the annotation of the different cell types composing the root (limited to clusters #0, 1 and 2)]. In response to this limitation in resolution, this reviewer requested the generation of single-cell/nucleus transcriptomic to enrich the spatial datasets and generate a real "single-cell-resolution" assessment of gene expression. Unfortunately, the authors did not conduct such an experiment to enrich their manuscript. While this reviewer agrees with the authors that "the most frequently employed integration method involves leveraging gene expression site information from spatial transcriptomics to annotate the single-cell transcriptomes", this reviewer believes that the two technologies support each other. Therefore, in this manuscript, the use of single-cell or single-nucleus RNA-seq experiments will be used to support the interpretation of the spatial transcriptomic analysis.

Minor comment: Line 67: The authors stated that their 10 dpi nodules are mature nodules. This is by far not a correct statement that can mislead the interpretation of the data.

Reviewer #3 (Remarks to the Author):

Dear Editor,

I read the revised manuscript by Ye and colleagues. I found the manuscript to be much improved and has addressed most of my critical comments from my original review. However, I made a few notes that might be addressed in the revision that could further improve the manuscript.

1. The authors use two versions of the Monocle software, seemingly to have orthogonal approaches to support their pseudotime trajectories. I would recommend selecting one and (if needed) using an alternative approach like Slingshot or CellRank to compute trajectories for comparison.
2. In figure 2 (infection zone subclustering and analysis), the UMAP in panel (a) includes a large number of gray-colored bins that are not described. From which major cluster are these derived, and what is the authors interpretation?
3. The dual-luciferase data showing the action of LtNLP3 (Extended Fig. 5b) on putative targets lacks a non-induced control
4. The bin-level metadata and analysis code is not available, so it will be impossible for someone to reproduce analysis. I'd recommend making code available on GitHub and creating a supplemental table with bin-level metadata, along with a bin x gene processed data table. If this is included in the STT0000041 repository, it was not open to the public so could not be reviewed.

RESPONSE TO REVIEWER COMMENTS

Reviewer #2 (Remarks to the Author):

1. This reviewer wants to thank the authors for answering the comments and for sharing Supplemental Table 1. This reviewer understands that the authors analyzed 172 million of unique reads (less than 1.5% of the 11.69 giga sequenced reads which is surprising to this reviewer) across 19,872 bins. This reviewer needs more information to better assess the quality of the data including the average numbers of expressed genes and UMI per bin.

Response: Thanks for this suggestion. In sequencing-based spatial transcriptomic methods, a thin tissue slice typically ranging from 10 to 20 μm is employed to extract mRNA. Given the trace quantity of mRNA present, amplification steps are essential, often involving 15 rounds of PCR in Stereo-seq library preparation¹. This amplification, however, leads to a notable proportion of duplicated reads. To guarantee accurate quantification, we implemented a filtration step to exclude these duplicated reads. It is noteworthy that this duplication issue and the corresponding filtration strategy are common across various spatial transcriptomic technologies¹⁻³. Moreover, the reads that failed to be annotated were primarily rRNA, which constitute the majority of the total RNA content. We observed that these rRNA were likely captured by the DNB due to the presence of several consecutive 'A' motifs in their sequences. Notably, our study incorporated bulk transcriptomics, which does not have the issue of limited sample contents. The resulting data covers approximately 80% of the genes in *L. japonicus*, specifically 80.69% at 4 dpi, 80.79% at 5 dpi, and 80.78% at 6 dpi. This closely aligns with our spatial transcriptomic data, which also covers approximately 80% of the genes. This congruency validates the quality of our spatial transcriptomic data. To enhance clarity, we have appended an illustration to Supplementary Table 1.

We analyzed the numbers of UMI and genes per bin, as illustrated in Supplementary Fig. 2a. Leveraging the spatial information, we discovered that the mRNA content exhibits distinct developmental and region-specific patterns (Supplementary Fig. 2b). We have added the information of average UMI/genes counts per bin to Supplementary Table 1.

2. Besides, this reviewer does not understand the rationale for conducting a bulk transcriptomic experiment in response to reviewer #2's first comment. The authors seem to agree that the used spatial transcriptomics is not a technology that supports single-cell resolution transcriptomic analyses. Considering that the present study generated 19,872 bins across 72 cross-sections, it means that each cross-section is composed of an average of 267 bins of $36 \times 36 \mu\text{m}^2$ to $60 \times 60 \mu\text{m}^2$. This reviewer assumes that many bins of such area cannot cover one cell as suggested by the authors (lines 611-612).

Response: Thanks for this suggestion. The purpose of conducting bulk transcriptomic experiments is to validate the precision of our spatial transcriptomic analysis in defining the developmental stages of the infection zone. We have revised this section to enhance

its comprehensiveness (lines 149-156). This issue is closely related to the subsequent one, which will be addressed in more detail.

Before determining the parameters for spatial transcriptomic analysis, we measured the length of *L. japonicus* nodule cells. Given the significance of the (pre-)infection zone, the chosen parameters primarily focused on this region, particularly the infected cells where symbiosis occurs. We have added this information to Supplementary Table 1. Before 6 dpi, the cells belonging to this region measured approximately 30 μm . However, the infected cells underwent a remarkable enlargement thereafter, reaching a size of around 60 μm . Therefore, we selected bin parameters of 36×36 and 60×60 μm^2 . Since other types of cells are generally smaller than the infected cells, especially during the late developmental stages, it is conceivable that certain bins may accommodate multiple cells.

3. The lack of resolution might affect the interpretation of the data. For instance, this reviewer is not certain that this study is truly “an accurate, tissue-resolution molecular map” as claimed by the authors because each bin might not reflect a group of cells sharing similar functions [e.g., this reviewer did not notice the group of the uninfected cells of the infection zone of the *L. japonicus* nodule or the annotation of the different cell types composing the root (limited to clusters #0, 1 and 2)]. In response to this limitation in resolution, this reviewer requested the generation of single-cell/nucleus transcriptomic to enrich the spatial datasets and generate a real “single-cell-resolution” assessment of gene expression. Unfortunately, the authors did not conduct such an experiment to enrich their manuscript. While this reviewer agrees with the authors that “the most frequently employed integration method involves leveraging gene expression site information from spatial transcriptomics to annotate the single-cell transcriptomes”, this reviewer believes that the two technologies support each other. Therefore, in this manuscript, the use of single-cell or single-nucleus RNA-seq experiments will be used to support the interpretation of the spatial transcriptomic analysis.

Response: Thank you for your suggestion. We have acknowledged that the tissue-resolution is not attainable in certain regions, and we have proofread our manuscript carefully to prevent any miswriting. Our goal aligns with yours, as we aim to ensure that this study yields meaningful insights and valuable data.

The development of root nodule is an intricate process, further compounded by the challenges of their subterranean formation, which complicates the sampling. As a result, a comprehensive transcriptomic atlas for determinate nodule development has remained elusive^{4,5}. Before embarking on this research, we conducted a thorough analysis of the strengths and weaknesses of spatial transcriptomics and single-cell/nucleus transcriptomics, ultimately concluding that spatial transcriptomics met our objectives. While this cutting-edge technology has limitations such as constricted sample volume and relatively low resolution⁶, it offers rapid material processing that preserves transcriptome authenticity. Additionally, when combined with morphological and anatomical observations, it enables more precise outcomes, particularly beneficial in plant studies with limited marker genes^{7,8}. These characteristics are advantageous for investigating nodule development.

Although single-cell RNA-seq has been widely utilized in plant research, the extensive dissociation process required due to plant cell walls impacts the transcriptome quality⁹⁻¹². Additionally, the infected cells of root nodules exhibit notable size disparities during development, which further compromises data quality¹³⁻¹⁵. Apart from the potential issue of low-quality data in single-cell RNA-seq, integrating it with spatial transcriptomic analysis raises concerns about batch effects. Despite available algorithms to mitigate this effect, it remains a challenge in both methodologies^{16,17}. Thereby, successful integration analyses of these two data types have been reported only in several animal research^{18,19}, with spatial transcriptomics primarily used to obtain gene expression location information, serving as a complementary tool to single-cell analysis²⁰. A recent plant study has exemplified such an application, focusing on single-cell analysis, but underutilized spatial transcriptomic data²¹.

Collectively, we have chosen to employ spatial transcriptomics, leveraging our expertise in Stereo-seq developed by BGI company¹, to advance our understanding of nodule development. To address limited sample contents, we secured sufficient biological replicates for robust statistical analysis. Our research delineated the infection zone's developmental trajectory, pinpointing infection initiation at 6 dpi in *L. japonicus* nodules, and providing candidate genes for investigating this crucial transition stage. As previous studies emphasized infection as the pivotal event triggering transcriptional changes^{22,23}, we performed bulk transcriptomics, and validated that transcriptome shifts significantly at 6 dpi compared to 4 and 5 dpi (Fig. 2i and Supplementary Fig. 4i-4l), reinforcing the credibility of our findings.

To overcome low resolution, BGI has developed Stereo-seq with cell-resolution precision, promising diverse cell type identification within the same tissue^{1,24,25}. We are committed to enhancing its application in plant research^{2,26}, confident that cell-resolution spatial transcriptomics will become a convenient tool for exploring intricate biological processes, including nodule development.

4. Minor comment: Line 67: The authors stated that their 10 dpi nodules are mature nodules. This is by far not a correct statement that can mislead the interpretation of the data.

Response: Thanks for this suggestion. By integrating our findings with prior research²⁷⁻²⁹, we have substantiated the maturity of nodules at 10 dpi, evidenced by morphological and anatomical observations (Supplementary Fig. 1a-1d). Furthermore, our spatial data has uncovered a notable decrease in UMI and gene counts during this phase (Supplementary Fig. 2a and 2b). This observation is consistent with earlier studies, implying a potential decline in mRNA content during late developmental stages^{1,30,31}. We have refined the sentence to offer a more accurate description (lines 69-71).

Reviewer #3 (Remarks to the Author):

Dear Editor,

I read the revised manuscript by Ye and colleagues. I found the manuscript to be much improved and has addressed most of my critical comments from my original review. However, I made a few notes that might be addressed in the revision that could further improve the manuscript.

1. The authors use two versions of the Monocle software, seemingly to have orthogonal approaches to support their pseudotime trajectories. I would recommend selecting one and (if needed) using an alternative approach like Slingshot or CellRank to compute trajectories for comparison.

Response: Thank you for your valuable suggestion. Given the distinct algorithms employed by Monocle 2 and Monocle 3, we have incorporated both tools to reinforce our findings^{15,24,30}. Furthermore, we have followed your recommendation and included the Slingshot analysis, yielding consistent trajectories that validate our interpretation (Fig 2d and 3c).

2. In figure 2 (infection zone subclustering and analysis), the UMAP in panel (a) includes a large number of gray-colored bins that are not described. From which major cluster are these derived, and what is the authors interpretation?

Response: Thanks for this suggestion. In this section, we have re-clustered clusters 5, 8, and 9 to refine the infection zone's developmental trajectories. Fig. 2a illustrates the dissection of cluster 9 into three distinct subclusters, highlighting that post-infection differentiation involves multiple distinct developmental stages. Additionally, the spatial arrangement of these subclusters within the UMAP plot of major clusters correlates with their developmental direction. To improve clarity, we have labeled all unrelated clusters in gray. We have made necessary adjustments to both the figure and its legend for better comprehension.

3. The dual-luciferase data showing the action of LtNLP3 (Supplementary Fig. 5b) on putative targets lacks a non-induced control.

Response: Thank you for your valuable suggestion. The transient activation assays, including the dual luciferase experiment, are designed to verify whether specific effectors, typically transcription factors, regulate potential targets. In such experiments, proteins lacking regulatory activity, such as GFP, are frequently employed as controls³²⁻³⁴. In our study, we also employed GFP as the control for LjNLP3. We have made necessary adjustments to Supplementary Fig. 5b and its legend to improve clarity.

4. The bin-level metadata and analysis code are not available, so it will be impossible for someone to reproduce analysis. I'd recommend making code available on GitHub and creating a supplemental table with bin-level metadata, along with a bin x gene processed data table. If this is included in the STT0000041 repository, it was not open to the public so could not be reviewed.

Response: Thank you for your valuable suggestion. The STT0000041 repository encompasses bin-level metadata and bin×gene processed data. We will release this repository promptly following the publication of our work, ensuring that everyone has

access to the comprehensive data.

References

- 1 Chen, A. *et al.* Spatiotemporal transcriptomic atlas of mouse organogenesis using DNA nanoball-patterned arrays. *Cell* **185**, 1777-1792 e1721, doi:10.1016/j.cell.2022.04.003 (2022).
- 2 Song, X. *et al.* Spatial transcriptomics reveals light-induced chlorenchyma cells involved in promoting shoot regeneration in tomato callus. *Proc Natl Acad Sci U S A* **120**, e2310163120, doi:10.1073/pnas.2310163120 (2023).
- 3 Liu, C. *et al.* A spatiotemporal atlas of organogenesis in the development of orchid flowers. *Nucleic Acids Res* **50**, 9724-9737, doi:10.1093/nar/gkac773 (2022).
- 4 Mun, T., Bachmann, A., Gupta, V., Stougaard, J. & Andersen, S. U. Lotus Base: An integrated information portal for the model legume *Lotus japonicus*. *Sci. Rep.* **6**, 39447, doi:10.1038/srep39447 (2016).
- 5 Kelly, S., Mun, T., Stougaard, J., Ben, C. & Andersen, S. U. Distinct Transcriptomic Responses to a Spectrum of Bacteria Ranging From Symbiotic to Pathogenic. *Frontiers in Plant Science* **9**, doi:10.3389/fpls.2018.01218 (2018).
- 6 Zormpas, E., Queen, R., Comber, A. & Cockell, S. J. Mapping the transcriptome Realizing the full potential of spatial data analysis. *Cell* **186**, 5677-5689, doi:10.1016/j.cell.2023.11.003 (2023).
- 7 Yin, R., Xia, K. & Xu, X. Spatial transcriptomics drives a new era in plant research. *Plant J* **116**, 1571-1581, doi:10.1111/tpj.16437 (2023).
- 8 Rhaman, M. S., Ali, M., Ye, W. & Li, B. Opportunities and Challenges in Advancing Plant Research with Single-cell Omics. *Genomics, Proteomics & Bioinformatics*, doi:10.1093/gpbjnl/qzae026 (2024).
- 9 Seyfferth, C. *et al.* Advances and Opportunities in Single-Cell Transcriptomics for Plant Research. *Annu Rev Plant Biol* **72**, 847-866, doi:10.1146/annurev-arplant-081720-010120 (2021).
- 10 Satterlee, J. W., Strable, J. & Scanlon, M. J. Plant stem-cell organization and differentiation at single-cell resolution. *Proc Natl Acad Sci U S A* **117**, 33689-33699, doi:10.1073/pnas.2018788117 (2020).
- 11 Shaw, R., Tian, X. & Xu, J. Single-Cell Transcriptome Analysis in Plants: Advances and Challenges. *Molecular Plant* **14**, 115-126, doi:10.1016/j.molp.2020.10.012 (2021).
- 12 van den Brink, S. C. *et al.* Single-cell sequencing reveals dissociation-induced gene expression in tissue subpopulations. *Nat Methods* **14**, 935-936, doi:10.1038/nmeth.4437 (2017).
- 13 Libault, M. Transcriptional Reprogramming of Legume Genomes: Perspective and Challenges Associated With Single-Cell and Single Cell-Type Approaches During Nodule Development. *Front Plant Sci* **9**, 1600, doi:10.3389/fpls.2018.01600 (2018).
- 14 Wang, L. L. *et al.* Single cell-type transcriptome profiling reveals genes that promote nitrogen fixation in the infected and uninfected cells of legume nodules. *Plant Biotechnol J* **20**, 616-618, doi:10.1111/pbi.13778 (2022).
- 15 Ye, Q. *et al.* Differentiation trajectories and biofunctions of symbiotic and un-symbiotic fate cells in root nodules of *Medicago truncatula*. *Mol. Plant* **15**, 1852-1867, doi:10.1016/j.molp.2022.10.019 (2022).
- 16 Tran, H. T. N. *et al.* A benchmark of batch-effect correction methods for single-cell RNA sequencing data. *Genome Biol* **21**, 12, doi:10.1186/s13059-019-1850-9 (2020).

- 17 Yu, X., Xu, X., Zhang, J. & Li, X. Batch alignment of single-cell transcriptomics data using
deep metric learning. *Nat Commun* **14**, 960, doi:10.1038/s41467-023-36635-5 (2023).
- 18 Moncada, R. *et al.* Integrating microarray-based spatial transcriptomics and single-cell RNA-
seq reveals tissue architecture in pancreatic ductal adenocarcinomas. *Nature Biotechnology* **38**,
333-+, doi:10.1038/s41587-019-0392-8 (2020).
- 19 Longo, S. K., Guo, M. G., Ji, A. L. & Khavari, P. A. Integrating single-cell and spatial
transcriptomics to elucidate intercellular tissue dynamics. *Nat Rev Genet* **22**, 627-644,
doi:10.1038/s41576-021-00370-8 (2021).
- 20 A call for spatial omics submissions. *Nat Genet* **56**, 1-1, doi:10.1038/s41588-023-01621-6
(2024).
- 21 Liu, Z. *et al.* Integrated single-nucleus and spatial transcriptomics captures transitional states in
soybean nodule maturation. *Nature Plants*, doi:10.1038/s41477-023-01387-z (2023).
- 22 Mergaert, P., Kereszt, A. & Kondorosi, E. Gene expression in nitrogen-fixing symbiotic nodule
cells in *Medicago truncatula* and other nodulating plants. *Plant Cell* **32**, 42-68,
doi:10.1105/tpc.19.00494 (2020).
- 23 Hogslund, N. *et al.* Dissection of symbiosis and organ development by integrated transcriptome
analysis of mutant and wild-type plants. *Plos One* **4**, doi:10.1371/journal.pone.0006556 (2009).
- 24 Wei, X. Y. *et al.* Single-cell Stereo-seq reveals induced progenitor cells involved in axolotl brain
regeneration. *Science* **377**, 1062-+, doi:10.1126/science.abp9444 (2022).
- 25 Cheng, M. *et al.* Spatially resolved transcriptomics: a comprehensive review of their
technological advances, applications, and challenges. *J Genet Genomics* **50**, 625-640,
doi:10.1016/j.jgg.2023.03.011 (2023).
- 26 Xia, K. *et al.* The single-cell stereo-seq reveals region-specific cell subtypes and transcriptome
profiling in *Arabidopsis* leaves. *Dev. Cell* **57**, 1299-1310 e1294,
doi:10.1016/j.devcel.2022.04.011 (2022).
- 27 Szczyglowski, K. *et al.* Nodule organogenesis and symbiotic mutants of the model legume *Lotus*
japonicus. *Mol. Plant-Microbe Interact.* **11**, 684-697, doi:10.1094/mpmi.1998.11.7.684 (1998).
- 28 Guinel, F. C. Getting around the legume nodule: I. The structure of the peripheral zone in four
nodule types. *Botany* **87**, 1117-1138, doi:10.1139/b09-074 (2009).
- 29 Ferguson, B. J. *et al.* Molecular analysis of legume nodule development and autoregulation. *J.*
Integr. Plant Biol. **52**, 61-76, doi:10.1111/j.1744-7909.2010.00899.x (2010).
- 30 Cao, J. *et al.* The single-cell transcriptional landscape of mammalian organogenesis. *Nature* **566**,
496-502, doi:10.1038/s41586-019-0969-x (2019).
- 31 Mu, T. H. *et al.* Embryonic liver developmental trajectory revealed by single-cell RNA
sequencing in the *Foxa2* mouse. *Commun Biol* **3**, doi:10.1038/s42003-020-01364-8 (2020).
- 32 Wang, L. *et al.* Transcriptional regulation of strigolactone signalling in. *Nature* **583**, 277-+,
doi:10.1038/s41586-020-2382-x (2020).
- 33 Misawa, F. *et al.* Nitrate transport via NRT2.1 mediates NIN-LIKE PROTEIN-dependent
suppression of root nodulation in *Lotus japonicus*. *Plant Cell* **34**, 1844-1862,
doi:10.1093/plcell/koac046 (2022).
- 34 Fonouni-Farde, C. *et al.* DELLA-mediated gibberellin signalling regulates Nod factor signalling
and rhizobial infection. *Nat Commun* **7**, 12636, doi:10.1038/ncomms12636 (2016).

REVIEWERS' COMMENTS

Reviewer #2 (Remarks to the Author):

Review of the manuscript entitled "Mapping the Molecular Landscape of Lotus japonicus Nodule Organogenesis through Spatiotemporal Transcriptomics" by Ye et al.

This reviewer wants to thank the authors for sharing the details of their sequencing and mapping results. As described in more detail below, this reviewer noticed that the depth of the stereo-seq transcriptome is low, sometime very low when considering the 10 dpi nodule samples. This raises concerns about data quality and, a fortiori, their interpretation.

In their response, the authors mentioned that the thickness of the cross-section (10 to 20 μm) justifies the "trace quantity of mRNA" analyzed by RNA-seq. This reviewer disagrees with such an explanation. Researchers working on plant single-nucleus RNA-seq are now used to detect more than 1,000 expressed genes per nucleus. Compared to the Stereo-seq experiments conducted by the authors, the nucleus would represent a small volume compared to the volume of the cells analyzed by the authors (i.e., considering a nucleus of 10 μm of diameter, it will have a volume of 1,340 μm^3 vs. 26,000 μm^3 for a cross-section of 20 μm and a bin of 36 μm x 36 μm). Considering the depth of the sequencing which is very high and likely saturates the Stereo-seq libraries (see my previous review), this reviewer concludes that Stereo-seq experiments conducted by the authors only captured a small fraction of the cellular transcriptome.

Hence, the quality of the information shared in this study is limited for two reasons: 1- because it is not a single cell-resolution gene expression analysis (i.e., use of 36 μm x 36 μm or 58 μm x 58 μm bins) and 2- because it is not a deep transcriptomic atlas of the different cell-types composing the Lotus japonicus nodule. Therefore, this reviewer also disagrees with the authors how claimed that "BGI has developed Stereo-seq with cell-resolution precision". As a note, the authors are still sharing an ambiguous statement about the resolution of their work in lines 482-483).

To overcome the lack of resolution of Stereo-seq, the authors declined to generate a single-cell transcriptome of the Lotus nodule. The authors commented on the difficulty of digesting the cell wall and the potential bursting of the protoplasts when conducting a single-cell transcriptomic approach. This reviewer agrees with the authors on this specific point. However, this reviewer was surprised that the authors did not consider the single-nucleus approach. This is unfortunate because the inclusion of some "real" single cell dataset would have considerably strengthen the quality of the data.

The authors also missed an opportunity to demonstrate the quality of the information generated in their manuscript by comparing their analysis with previous tissue-resolution gene expression studies (e.g., 10.1111/pbi.13778; <https://doi.org/10.1038/s41467-023-42911-1>; 10.1094/MPMI-01-12-0011-R). Based on the studies above, the novelty of the data analyzed by the authors is unclear to this reviewer.

Reviewer #3 (Remarks to the Author):

In the revised submission, the authors have addressed my concerns.

Responses to referees' comments:

Referee #2 (Remarks to the Author):

This reviewer wants to thank the authors for sharing the details of their sequencing and mapping results. As described in more detail below, this reviewer noticed that the depth of the stereo-seq transcriptome is low, sometime very low when considering the 10 dpi nodule samples. This raises concerns about data quality and, a fortiori, their interpretation.

In their response, the authors mentioned that the thickness of the cross-section (10 to 20 μm) justifies the “trace quantity of mRNA” analyzed by RNA-seq. This reviewer disagrees with such an explanation. Researchers working on plant single-nucleus RNA-seq are now used to detect more than 1,000 expressed genes per nucleus. Compared to the Stereo-seq experiments conducted by the authors, the nucleus would represent a small volume compared to the volume of the cells analyzed by the authors (i.e., considering a nucleus of 10 μm of diameter, it will have a volume of 1,340 μm^3 vs. 26,000 μm^3 for a cross-section of 20 μm and a bin of 36 μm x 36 μm). Considering the depth of the sequencing which is very high and likely saturates the Stereo-seq libraries (see my previous review), this reviewer concludes that Stereo-seq experiments conducted by the authors only captured a small fraction of the cellular transcriptome.

Hence, the quality of the information shared in this study is limited for two reasons: 1- because it is not a single cell-resolution gene expression analysis (i.e., use of 36 μm x 36 μm or 58 μm x 58 μm bins) and 2- because it is not a deep transcriptomic atlas of the different cell-types composing the *Lotus japonicus* nodule. Therefore, this reviewer also disagrees with the authors how claimed that “BGI has developed Stereo-seq with cell-resolution precision”. As a note, the authors are still sharing an ambiguous statement about the resolution of their work in lines 482-483).

To overcome the lack of resolution of Stereo-seq, the authors declined to generate a single-cell transcriptome of the *Lotus* nodule. The authors commented on the difficulty of digesting the cell wall and the potential bursting of the protoplasts when conducting a single-cell transcriptomic approach. This reviewer agrees with the authors on this specific point. However, this reviewer was surprised that the authors did not consider the single-nucleus approach. This is unfortunate because the inclusion of some “real” single cell dataset would have considerably strengthen the quality of the data.

The authors also missed an opportunity to demonstrate the quality of the information generated in their manuscript by comparing their analysis with previous tissue-resolution gene expression studies (e.g., 10.1111/pbi.13778; <https://doi.org/10.1038/s41467-023-42911-1>; 10.1094/MPMI-01-12-0011-R). Based on the studies above, the novelty of the data analyzed by the authors is unclear to this reviewer.

Response: Thanks for this suggestion. Our spatial transcriptomic data reveals that developing nodules harbor significantly higher RNA contents compared to their neighboring root tissues, with the majority of genes detected per bin surpassing 1000 (Supplementary Fig. 2b). However, upon reaching maturity at 10 dpi, a notable decrease in RNA contents is observed, which aligns with several established patterns of organ development¹⁻³. It is noteworthy that genes such as *LjLbs* are more prominently expressed during this mature stage (Fig 2m), consistent with other nodulation studies^{4,5}, despite the reduced RNA contents. This underscores that the variations in RNA content are linked to nodule development, rather than data quality. Furthermore, as previously mentioned, our study incorporates bulk transcriptomics, and the similarity in gene coverage between these two data types further validates that the spatial transcriptome has captured a comprehensive transcriptome.

Hindered by challenges like sampling, previous transcriptomic investigations on determinate nodules have centered on specific developmental stage, mostly mature stage⁶⁻⁸. Leveraging spatial transcriptomics, we have conducted this pioneering study into the molecular development of determinate nodules, revealing valuable insights. To interpret our spatial transcriptomic data, we have referred to previous nodulation studies, including transcriptomic studies on several legumes⁹⁻¹¹. This comprehensive exploration provides molecular validation for past hypotheses (Fig. 3d, 3e and Supplementary Fig. 7a), and sheds light on potential candidate genes for further exploration (Fig. 2e and Supplementary Fig. 6f). Among these genes, we have conducted a thorough functional analysis of *LjNLP3*, emphasizing the advantages of our spatial transcriptomic data (Supplementary Fig. 5). Additionally, we aim to further investigate other candidate functional genes identified in this research, including *VPS13s*, which may provide insights into the endocytic processes that are critical for symbiosome maturation (Fig. 3h and 3i). We firmly believe that our data and findings are valuable for researchers in this field.

References

- 1 Cao, J. *et al.* The single-cell transcriptional landscape of mammalian organogenesis. *Nature* **566**, 496-502, doi:10.1038/s41586-019-0969-x (2019).
- 2 Mu, T. H. *et al.* Embryonic liver developmental trajectory revealed by single-cell RNA sequencing in the Foxa2 mouse. *Commun Biol* **3**, doi:10.1038/s42003-020-01364-8 (2020).
- 3 Chen, A. *et al.* Spatiotemporal transcriptomic atlas of mouse organogenesis using DNA nanoball-patterned arrays. *Cell* **185**, 1777-1792 e1721, doi:10.1016/j.cell.2022.04.003 (2022).
- 4 Downie, J. A. Legume haemoglobins: symbiotic nitrogen fixation needs bloody nodules. *Curr. Biol.* **15**, R196-198, doi:10.1016/j.cub.2005.03.007 (2005).
- 5 Larrainzar, E. *et al.* Hemoglobins in the legume-Rhizobium symbiosis. *New Phytol.* **228**, 472-484, doi:10.1111/nph.16673 (2020).
- 6 Liu, Z. *et al.* Integrated single-nucleus and spatial transcriptomics captures transitional states in soybean nodule maturation. *Nature Plants*, doi:10.1038/s41477-023-01387-z (2023).
- 7 Sun, B. C. *et al.* A high-resolution transcriptomic atlas depicting nitrogen fixation and nodule development in soybean. *J. Integr. Plant Biol.* **65**, 1536-1552, doi:10.1111/jipb.13495 (2023).
- 8 Wang, L. L. *et al.* Single cell-type transcriptome profiling reveals genes that promote nitrogen fixation in the infected and uninfected cells of legume nodules. *Plant Biotechnol J* **20**, 616-618, doi:10.1111/pbi.13778 (2022).
- 9 Limpens, E. *et al.* Cell- and tissue-specific transcriptome analyses of *Medicago truncatula* root nodules. *Plos One* **8**, doi:10.1371/journal.pone.0064377 (2013).
- 10 Ye, Q. *et al.* Differentiation trajectories and biofunctions of symbiotic and un-symbiotic fate cells in root nodules of *Medicago truncatula*. *Mol. Plant* **15**, 1852-1867, doi:10.1016/j.molp.2022.10.019 (2022).
- 11 Hogslund, N. *et al.* Dissection of symbiosis and organ development by integrated transcriptome analysis of mutant and wild-type plants. *Plos One* **4**, doi:10.1371/journal.pone.0006556 (2009).